# The Effects of Lighting Problems on Eye Symptoms among Cleanroom Microscope Workers

**DOI:** 10.3390/ijerph16010101

**Published:** 2019-01-01

**Authors:** Kuan-Han Lin, Chien-Chia Su, Yen-Yuan Chen, Po-Ching Chu

**Affiliations:** 1Graduate Institute of Medical Education & Bioethics, National Taiwan University College of Medicine, #1, Rd. Ren-Ai Sec. 1, Taipei 10051, Taiwan; okonkwolin@gmail.com (K.-H.L.); chen.yenyuan@gmail.com (Y.-Y.C.); 2Department of Ophthalmology, National Taiwan University Hospital, #7, Chung-Shan South Road, Taipei 10002, Taiwan; kido0214@yahoo.com.tw; 3Department of Medical Education, National Taiwan University Hospital, #7, Chung-Shan South Road, Taipei 10002, Taiwan; 4Department of Environmental and Occupational Medicine, National Taiwan University College of Medicine, #1, Rd. Ren-Ai Sec. 1, Taipei 10051, Taiwan; 5Department of Environmental and Occupational Medicine, National Taiwan University Hospital, #7, Chung-Shan South Road, Taipei 10002, Taiwan

**Keywords:** visual health, lighting, microscope, cleanroom

## Abstract

The visual health of microscope workers is an important occupational health concern, and a previous study suggested an association between lighting problems (e.g., flashing light, insufficient lighting) and eye symptoms among cleanroom workers in the electronics industry. This study aimed to explore the association between eye symptoms and lighting problems, as well as light-related counteracting behaviors among microscope workers in the cleanroom environment. Ninety-one cleanroom workers aged 20 years or older were recruited from an electronics factory. The socio-demographic factors, work-related factors, eye symptoms, and lighting problems were assessed using a self-administered questionnaire. There were 92.3% female participants in this study. Among all participants, 41.8% and 63.7% had symptoms of dry eye and eye fatigue, respectively. The counteracting behaviors of needing to move closer (adjusted odds ratio (aOR) = 3.47, 95% CI = 1.11 to 10.88) was significantly associated with dry eye symptoms. Workers who were more experienced at the job (aOR = 1.03, 95% CI = 1.01 to 1.06) and had shorter break times (aOR = 0.94, 95% CI = 0.91 to 0.98) were more likely to have eye fatigue. As a result of these findings, this study suggests that good lighting and adequate break times are crucial to improve the visual health of cleanroom microscope workers.

## 1. Introduction

The electronics industry is estimated to have 18 million workers employed worldwide in 2010 [1], and the estimated growth of this industry was around 4% between 2016 and 2017 globally [2]. East Asian countries including Japan and Taiwan contribute to over 50% of the global export values. To ensure the quality of the products, small electronic components are generally manufactured in cleanrooms with controlled temperature, humidity, and air particulate matter concentration [3]. Special head-to-toe garments are required to reduce dust or lint exposure, but the garments may limit the field of view and restrict the range of movement, causing discomfort when the workers remain fully suited during the entire work shift. High prevalence of dry eye symptoms has been reported among cleanroom workers, and the symptoms may be reduced by allocating adequate working hours and wearing protective eyewear [4,5]. Common light hazards in the cleanroom environment include poor lighting, special lamp design (e.g., yellow light), high illuminance (e.g., microscope), and low illuminance (e.g., light-on test) [4]. These hazards may result in lighting problems, such as disability glare, discomfort glare, flicker, and veiling reflections, and cause eye discomfort [6,7,8]. Toomingas et al. indicated that non-optimal visual conditions (e.g., poor lighting, glare) was an independent risk factor for eye symptoms [9]. Although prior studies reported that dry eye symptoms at the workplace are associated with poor lighting, female gender, cleanroom, long employment duration, family history of atopic disease, and low humidity [5,10,11], the association between different lighting problems and eye symptoms among cleanroom workers is rarely addressed. Furthermore, dry eye syndrome may also be more prevalent in people of Asian and Hispanic origin [10,12].

Cleanroom workers are often required to manually inspect the products at short distances by eye or with microscopes. Small myopic shifts have been found in previous studies on near work and were suggested to produce eye fatigue [13,14]. The field of view under the microscope tends to have a much higher illuminance than the surrounding environment, and the large differences in illumination between the field of vision of the microscopes and the adjacent working areas may aggravate eye symptoms such as eye fatigue [15]. Indeed, a previous study reported that 59.6% of microscope users in medical laboratories had eye fatigue [16], and a case report describing a pathologist who had severe eye symptoms and myopic complications from occupational exposure to both low-intensity fluorescent light and high intensity light [17]. However, there is insufficient evidence to show if an increased risk of eye fatigue is associated with different lighting problems among cleanrooms workers using microscopes.

Previous studies have found that glare affects people with normal binocular vision and contributes to decreased productivity [18], increased blood flow in the trapezius muscle [19], and an increased blink rate [6]. There are several human adaptations or counteracting behaviors, including eyelid squinting, blinking, and changing postures, to reduce the negative effects of glare [6]. Changing postures and work at close distances could be seen as the behaviors to cope with glare and intensive near-visual work, respectively [6,20]. Although the visual health for cleanroom workers or microscope workers have been reported [5,16,21], the association between counteracting behaviors and eye symptoms among cleanroom workers is rarely addressed. Therefore, the aims of this study were to explore the association between eye symptoms and lighting problems, as well as light-related counteracting behaviors among cleanroom workers in an electronics factory.

## 2. Materials and Methods

This is a cross-sectional study conducted by the Department of Quality Control at an electronic manufacturing enterprise in Northern Taiwan in 2011. The main product of the enterprise was light-emitting diodes, and the type of activity for the study population was mainly visual inspection. All available participants at the enterprise were invited to participate in the study. Formal written instructions for the study were posted in the workplace, and a research staff member presented a verbal briefing based on a written script to the participants prior to the distribution of the questionnaires. The participants were given the opportunity to decline participation or to withdraw at any time. Privacy was guaranteed during the study, and the study was anonymous.

A total of 135 participants were invited. The inclusion criteria are age 20 or older and microscope use of longer than 4 h per working day in the cleanroom. The relative humidity of the cleanroom was maintained at 55 ± 5%, and the workers performed the tasks with an environment illumination of about 500 Lux. A self-administered questionnaire with a cover letter explaining the purpose of our study was distributed to all workers’ mailboxes in the industry. Completion and return of the questionnaires to the infirmary of the company was considered the workers’ consent to participate. This study was approved by the Research Ethics Committee in National Taiwan University Hospital (201708044RINC).

The questionnaire collected demographic data, eye symptoms (e.g., dry eye and eye fatigue), non-eye related symptoms, work-related factors, lighting problems, and counteracting behaviors for lighting problems. The demographic data included age, gender, and history of myopia. Questions on eye symptoms were adapted from the Health Aspect of Lighting at Work by Health and Safety Executive [15]. In this study, we defined symptoms of dry eye as the presence of either dryness or irritation according to previous literature [22]. Non-eye-related symptoms including physical fatigue, neck pain, and shoulder pain were surveyed. We also assessed the work-related factors of shift type, experience at the job, working time, break time, duration of visually demanding tasks per working day, duration spent on display screens per working day, and the shortest distance between eyes and objects. With regard to lighting problems, we assessed the perceptions of disability glare (direct interference with vision), discomfort glare (not directly impaired but causing discomfort, annoyance, irritability, or distraction), requirement of color discrimination, flicker, and veiling reflections. One of the adaptations is counteracting behaviors like changing posture. In the study, needing to move closer and frequent shifting to view from different angles were chosen as counteracting behaviors for lighting problems.

All statistical analyses were conducted using SAS 9.4 (SAS Institute Inc., Cary, NC, USA). All data were expressed as frequency (percentage), mean ± S.D., or median (interquartile range). The differences in the distribution of demographic characteristics, work-related factors, lighting problems, and counteracting behaviors between workers with and without symptoms were examined using Student’s *t*-test, a chi-square test, and a Mann-Whitney U test, respectively. Any factors having a significant difference in the univariable test were selected as candidates for the univariable logistic regression analyses. Univariable logistic regression analyses were performed to examine the potential effect of work-related factors, lighting problems, and counteracting behaviors on symptoms of dry eye or eye fatigue. Multivariable logistic regression analyses were then used to adjust for age, gender, and other significant factors from the univariable logistic regression analyses. The calibration of the model was examined using the Hosmer and Lemeshow Goodness-of-fit test [23]. A *p*-value of less than or equal to 0.05 was considered statistically significant.

## 3. Results

A total of 91 microscope workers (response rate: 67.4%) completed the study. Among them, the majority were females (92.3%), and the average age was 31.5 years old (SD = 5.6). There were 54.9% who had a history of myopia. Thirty-four workers (37.4%) were on the night shift, and the median experience at the job was 14 months (interquartile range = 7–54.3). Workers spent 9.4 h each day staring at the screen on average, and the average shortest distance between eyes and objects was about 30 cm (Table 1). Among the 91 workers, the most prevalent eye symptom was eye fatigue (63.7%), followed by symptoms of dry eye (41.8%), eye itch (22.0%), and blurred vision or sense of oppression (13.2% each). Among non-eye related symptoms, the most frequently reported one was physical fatigue (38.5%), followed by shoulder pain (37.4%) and neck pain (30.8%). Workers with dry eye symptoms were more likely to have a history of myopia (*p* < 0.01), have physical fatigue (*p* < 0.01), have shoulder pain (*p* < 0.01), and have neck pain (*p* = 0.01) compared with those without dry eye symptoms. On the other hand, workers with eye fatigue were more likely to be female (*p* = 0.04), have long experience at the job (*p* < 0.01), have physical fatigue (*p* < 0.01), have shoulder pain (*p* < 0.01), have neck pain (*p* < 0.01), and have short break times (*p* = 0.01) when compared with those without eye fatigue (Table 1).

Regarding the lighting problems, the most frequently declared factor was the requirement of color discrimination (52.7%), but it was not associated with either dry eye or eye fatigue symptoms (Table 2). Overall, 15.4% reported discomfort glare. More workers with dry eye symptoms reported discomfort glare when compared with those without dry eye symptoms (23.7% vs. 9.4%), although the difference was insignificant (*p* = 0.06). The other reported lighting problems were not significantly different between workers with and without dry eye symptoms (Table 2). Regarding the counteracting behaviors, 29.7% reported that they need to move closer during work, and 25.3% needed to shift frequently to view from various angles. When compared with the workers without dry eye symptoms, more of those with dry eye symptoms had to move closer (*p* < 0.01) or shift frequently (*p* = 0.03). On the other hand, workers with eye fatigue were more likely to move closer compared with those without eye fatigue (*p* = 0.02).

The results of multivariable logistic regression analyses for factors associated with dry eye and eye fatigue symptoms are shown in Table 3 and Table 4, respectively. Needing to move closer was associated with dry eye symptoms after adjusting for sex, age, history of myopia, and frequent shifting to view from different angles (adjusted odds ratio (aOR) = 3.47, 95% CI = 1.11–10.88) (Table 3). Table 4 showed that experience at the job (aOR = 1.03, 95% CI = 1.01–1.06) was associated with eye fatigue and break time (aOR = 0.94, 95% CI = 0.91–0.98) had a protective effect on eye fatigue after adjusting for sex, age, break time, and needing to move closer. The *p*-values of the Hosmer and Lemeshow goodness-of-fit test were 0.70 and 0.81 for the two models, respectively, indicating good fitness of the models.

## 4. Discussion

The present study explored the effect of lighting problems and workers’ counteracting behaviors on eye symptoms among a special occupational population, which is the cleanroom microscope workers. Cleanroom microscope workers in the electronics industry are required to perform visually demanding work, but no previous study, to our knowledge, has explored the effect of lighting problems on different eye symptoms among these workers. We found that 41.8% and 63.7% of workers had symptoms of dry eye and eye fatigue, respectively. More workers with dry eye symptoms reported disability glare and discomfort glare than those without dry eye symptoms (23.7% vs. 13.2%; 23.7% vs. 9.4%, respectively). After adjusting for other factors, the counteracting behaviors of needing to move closer were significantly associated with dry eye symptoms. Moreover, experience at the job was a risk factor and break time was a protective factor of eye fatigue after adjusting for other factors.

Some studies that examine the association between work-related factors and dry eye symptoms [10,11] report that poor lighting, low humidity, and employment duration were considered as risk factors of dry eye in cleanroom [5]. The present study found that workers with dry eye symptoms reported a high prevalence of disability and discomfort glare, compared with those without dry eye symptoms. One possible reason accounting for the association between glare and dry eye symptoms is that glare put additional stress on the visual system [6]. Although glare has been reported to increase blink rates [6], subjects viewing electronic displays have a higher prevalence of incomplete blinks (occurring when the upper eyelid is unsuccessful in cover the entire corneal surface) [24], which may cause dry eye symptoms due to significant tear evaporation and tear break up [25]. Furthermore, the cleanroom microscope workers had prolonged high demand in performing visual inspections, which is a task with a high cognitive demand. Rosenfield et al. have found that high cognitive demand results in a significant reduction in mean blink rate [24], and Li et al. indicated that while eyes are focused on close objects, the number of blinks is decreased [26]. Taken together, we speculate that dry eye among these cleanroom workers may be associated with glare through changes in blink rates. The association between counteracting behaviors and dry eye symptoms among cleanroom workers is rarely addressed, and the present study further found that the counteracting behavior of needing to move closer was significantly associated with dry eye symptoms. It is possible that workers exposed to glare had a high prevalence of dry eye symptoms (Table 2), and were more likely to change posture to reduce glare [6,20]. During near work, counteracting behaviors, such as changing posture and shielding the eyes from the light source, are often used to reduce glare [20]. Changing posture and work at a close distance could be seen as human adaptations to cope with glare and extensive near work, respectively, and the common ways of changing posture was to bend the head forward, probably to keep excessive light from going into the eyes [6,20]. Furthermore, Li et al. indicated that near work combined with long-term use of video display terminals was associated with dry eye disease [26]. Therefore, prolonged exposure to glare and near work may give rise to negative visual health, such as dry eye symptoms [8,9,27]. 

In the present study, 63.7% of the workers had eye fatigue, and the average experience at the job was 14 months. Our findings showed that experience at the job was a risk factor associated with eye fatigue. Our finding is consistent with Su et al. [28] showing that longer employment duration was associated with increased risk of eye fatigue among quality control workers at light bulb check stations. Regarding break time and eye fatigue, a previous study found that working for more than six hours in front of the computer had a negative impact on eye fatigue [29], and frequent short breaks are an effective way to decrease eye fatigue [30]. Cheu indicated that looking at an object away from the screen every half an hour is sufficient for preventing eye fatigue [31]. Similar to the above studies, we also found that break time was a protective factor of eye fatigue. In addition to experience at the job and break time, differences in illuminance between the work area (local high lighting in the microscope) [17] and the adjacent area (systemic lighting in the working station) may cause visual discomfort [15]. Although the present study did not find an association between eye fatigue and lighting problems, future studies may focus on measuring the differences in illuminance to evaluate the contribution to eye fatigue among cleanroom microscope workers.

There were several limitations that should be noted. First, this is a study in a single facility and the generalizability of this study requires further evaluation. Second, this cross-sectional study limits the inference of causal relationships and can only determine the association between lighting problems and eye symptoms. Third, other potential confounding variables for symptoms of dry eye and eye fatigue, such as smoking and work stress, are not included in the multivariable logistic regression model due to lack of information. Furthermore, occupational eye injuries (e.g., ocular trauma due to small objects or chemical agents) [32,33,34] and exposure to solar radiation [35,36] were not considered in the present study. For example, Gobba et al. found that the annual frequency of work-related eye injuries for the computer and electronic industry was 11.3‰ [34]. Fourth, the small sample size may compromise the identification of potential associations between eye symptoms and different lighting problems. Lastly, the study population were predominantly young female workers, and the findings of the present study may not be generalized to other populations. Further studies are needed to include more cleanroom microscope workers of different gender and age groups to increase the understanding of this important public health issue concerning the large number of electronic workers worldwide.

## 5. Conclusions

We found that the counteracting behaviors of needing to move closer to cope with lighting problems was positively associated with dry eye symptoms in the cleanroom microscope workers after adjusting for sex, age, and history of myopia. Workers with more experience at the job and having shorter break times were more prone to develop eye fatigue. The results call attention to the importance of creating sustainable working conditions [9], including proper lighting, ergonomic workplace design, and adequate break time for cleanroom microscope workers. Further research may focus on objective estimates of different lighting effects such as glare and illuminance to determine whether they have an independent influence on eye symptoms among cleanroom workers. Development of a program for the early detection and prevention of eye symptoms in the working environment is also warranted.

## Figures and Tables

**Table 1 ijerph-16-00101-t001:** Distribution of demographic characteristics and work-related factors, stratified by symptoms of dry eye and eye fatigue.

Variables	Total (*N* = 91)	Without Dry Eye Symptoms (*N* = 53)	With Dry Eye Symptoms (*N* = 38)	*p*-Value	Without Eye Fatigue Symptom (*N* = 33)	With Eye Fatigue Symptom (*N* = 58)	*p*-Value
**Demographic Characteristics**							
Sex (female)	84 (92.3%)	47 (88.7%)	37 (97.4%)	0.13	28 (84.8%)	56 (96.6%)	0.04
Age (years)	31.5 ± 5.6	31.5 ± 6.1	31.4 ± 4.9	0.92	30.8 ± 5.6	31.9 ± 5.6	0.37
	(19–43)	(19–43)	(19–40)		(19–41)	(19–43)	
History of myopia	50 (54.9%)	23 (43.4%)	27 (71.1%)	<0.01	14 (42.4%)	36 (62.1%)	0.07
**Non-Eye Related Factors**							
Physical fatigue	35 (38.5%)	14 (26.4%)	21 (55.3%)	<0.01	4 (12.1%)	31 (53.4%)	<0.01
Shoulder pain	34 (37.4%)	13 (24.5%)	21 (55.3%)	<0.01	5 (15.2%)	29 (50.5%)	<0.01
Neck pain	28 (30.8%)	11 (20.8%)	17 (44.7%)	0.01	4 (12.1%)	24 (41.4%)	<0.01
**Work-Related Factors**							
Night shift	34 (37.4%)	20 (37.7%)	14 (36.8%)	0.93	16 (48.5%)	18 (31%)	0.1
Experience at the job (months)	14 (7–54.3)	11 (6–52.8)	23 (11–60.3)	0.06	8 (4–15)	29 (8–73)	<0.01 ^a^
Working time (minutes)	165.1 ± 60.8	168.4 ± 67.7	161.5 ± 52.8	0.63	171.7 ± 78.6	162.0 ± 51.0	0.59
Break time (minutes)	15.9 ± 16.7	18.0 ± 19.0	13.5 ± 13.6	0.26	24.8 ± 22.2	11.7 ± 11.4	0.01
Duration of visually demanding tasks per day (hours)	10.2 ± 1.6	10.4 ± 1.4	10.1 ± 1.7	0.34	10.6 ± 1.0	10.0 ± 1.8	0.07
Duration spent on display screens per day (hours)	9.4 ± 3.0	9.5 ± 3.1	9.4 ± 3.0	0.88	9.3 ± 3.4	9.5 ± 2.8	0.82
Shortest distance between eyes and objects (cm)	29.7 ± 13.7	30.8 ± 14.5	28.4 ± 12.6	0.43	32.3 ± 14.9	28.5 ± 13.1	0.24

Data are presented as number (%), mean ± standard deviation, or median (interquartile range). Note for pairwise deletion of missing values. ^a^ Mann-Whitney U test.

**Table 2 ijerph-16-00101-t002:** Distribution of lighting problems and counteracting behaviors, stratified by symptoms of dry eye and eye fatigue.

Variables	Total (*N* = 91)	Without Dry Eye Symptoms (*N* = 53)	With Dry Eye Symptoms (*N* = 38)	*p*-Value	Without Eye Fatigue Symptom (*N* = 33)	With Eye Fatigue Symptom (*N* = 58)	*p*-Value
**Lighting Problems**							
Disability glare	16 (17.6%)	7 (13.2%)	9 (23.7%)	0.20	6 (18.2%)	10 (17.2%)	0.91
Discomfort glare	14 (15.4%)	5 (9.4%)	9 (23.7%)	0.06	3 (9.1%)	11 (19.0%)	0.21
Required color discrimination	48 (52.7%)	25 (47.2%)	23 (60.5%)	0.21	15 (45.4%)	33 (56.9%)	0.29
Flicker	16 (17.6%)	7 (13.2%)	9 (23.7%)	0.20	6 (18.2%)	10 (17.2%)	0.91
Veiling reflections	15 (16.5%)	7 (13.2%)	8 (21.1%)	0.32	4 (12.1%)	11 (19.0%)	0.40
**Counteracting Behaviors**							
Needing to move closer	27 (29.7)	8 (15.1%)	19 (50.0%)	<0.01	5 (15.1%)	22 (37.9%)	0.02
Frequent shifting to view from different angles	23 (25.3)	9 (17.0%)	14 (36.8%)	0.03	8 (24.2%)	15 (25.9%)	0.86

Data are presented as number (%). Note for pairwise deletion of missing values.

**Table 3 ijerph-16-00101-t003:** Univariate and multivariate logistic regression analysis of factors influencing dry eye symptoms.

Variables	Crude OR (95% C.I.)	*p*-Value	aOR (95% C.I.) ^a^	*p*-Value
**Demographic characteristics**				
Sex				
Female	1.0	-	1.0	-
Male	0.21 (0.02–1.84)	0.16	0.30 (0.03–2.95)	0.30
Age (years)	1.00 (0.92–1.07)	0.92	0.99 (0.91–1.08)	0.82
History of myopia				
No	1.0	-	1.0	-
Yes	3.20 (1.32–7.77)	0.01	2.40 (0.90–6.37)	0.08
**Counteracting behaviors**				
Needing to move closer				
No	1.0	-	1.0	-
Yes	5.63 (2.10–15.06)	<0.01	3.47 (1.11–10.88)	0.03
Frequent shifting to view from different angles				
No	1.0	-	1.0	-
Yes	2.85 (1.08–7.55)	0.04	1.57 (0.47–5.26)	0.47

^a^ aOR: adjusted odds ratio, Note for Pairwise deletion of missing values.

**Table 4 ijerph-16-00101-t004:** Univariate and multivariate logistic regression analysis of factors influencing eye fatigue.

Variables	Crude OR (95% C.I.)	*p*-Value	aOR (95% C.I.) ^a^	*p*-Value
**Demographic Characteristics**				
Sex				
Female	1.0	-	1.0	-
Male	0.20 (0.04–1.10)	0.06	0.48 (0.05–4.46)	0.52
Age (years)	1.04 (0.96–1.12)	0.36	1.02 (0.91–1.14)	0.77
**Work-Related Factors**				
Experience at the job (months)	1.02 (1.00–1.04)	0.01	1.03 (1.01–1.06)	<0.01
Break time (minutes)	0.96 (0.92–0.99)	<0.01	0.94 (0.91–0.98)	<0.01
**Counteracting Behaviors**				
Needing to move closer				
No	1.0	-	1.0	-
Yes	3.42 (1.15–10.17)	0.03	3.46 (0.83–14.35)	0.09

^a^ aOR: adjusted odds ratio, Note for Pairwise deletion of missing values.

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
