# Peer review of "The Effects of Lighting Problems on Eye Symptoms among Cleanroom Microscope Workers"

_ijerph, 2019, doi:10.3390/ijerph16010101_

Round 1

Reviewer 1 Report

Overall an interesting and novel study which merits publication and dissemination to eye care professionals.  I have only a few minor comments, as laid out below:

Abstract:

Line 27: Please define OR as Odds Ratio

Introduction:

Lines 47-50: Dry Eye may also be more prevalent in people of Asian and Hispanic origin?  If the authors agree with this statement please add this information here.

Results:

Line 114: Please change the statement, 'among them, the majority was females' to 'among them, the majority were females'

Table 1: Please quote actual age range of each group, not just mean +/- SDM.

Statement in Discussion (Lines 209-210) in regard to 'preventing us from elucidating the effects of age and sex' seems at odds with end of first sentence in Conclusion section (Line 216) ('after adjusting for age and sex)

Author Response

Point 1: Abstract: Line 27: Please define OR as Odds Ratio. 

Response 1: Thank you for the comment. We have defined OR as Odds Ratio in the Abstract. Please see lines 28 of the revised file.

Point 2: Introduction: Lines 47-50: Dry Eye may also be more prevalent in people of Asian and Hispanic origin?  If the authors agree with this statement please add this information here.

Response 2: Thank you for the comment. We agree with this statement, and have add this information with new references in the Introduction. Please see lines 54-55 of the revised file.

Point 3: Results: Line 114: Please change the statement, 'among them, the majority was females' to 'among them, the majority were females'.

Response 3: Thank you for the comment. We have revised the statement in the Results. Please see lines 122-123 of the revised file.

Point 4: Results: Table 1: Please quote actual age range of each group, not just mean +/- SDM. 

Response 4: Thank you for the comment. We have added the actual age range of each group. Please see the revised Table 1 of the revised file.

Point 5: Statement in Discussion (Lines 209-210) in regard to 'preventing us from elucidating the effects of age and sex' seems at odds with end of first sentence in Conclusion section (Line 216) ('after adjusting for age and sex).

. 

Response 5: Thank you for the comment. Our original intention when we discuss the limitation on the study population being predominantly young female workers was to express that the findings of the present study may not be directly generalized to other populations. Thus, we have deleted 'preventing us from elucidating the effects of age and sex' and revised the statement in the Discussion. Please see lines 223-224 of the revised file.

Reviewer 2 Report

Dear Authors, the manuscript may provide an interesting overview of eye complaints among microscope workers in the electronic industry, nevertheless some major revisions are required before publication:

Abstract session:

Please re-write the sentence: "evidence suggests an association between lighting and eye symptoms among cleanroom workers in the electronics industry". "Evidence suggests" means nothing. Furthermore, I am not sure there is so much scientific literature on cleanroom workers, and the association for eye symptoms is not with lighting, but perhaps with insufficient lighting or with specific source of illumination (e.g. white/blu light, etc).

Introduction:

- Line 35-36: please find some more recent data: Authors are presenting data on occupation in the electronic industry related to approximately 10 years ago.

- Line 39: I suggest to (prevent) REDUCE dust or lint EXPOSURE 

- Line 41: High prevalence of dry eye SYMPTOMS

Materials and Methods

- Lines 70-71: some generic indications on the Country where the study was performed are required here, and I'd like to have more occupational details on the type of activity of the electronic enterprise to have a better idea of the work risks(types of materials produced, etc)

- "and a final of 91 participants (response rate was 67.4%) remained in the study population": this is a data for the results section

Results

I have no specific observations on the results of the study, that are almost a description of the eye concerns of the workers with respect to the workplace illumination and occupational activity, but these data are not sufficient to infere any conclusion on possible causal association as there is an inadequate consideration of possible confounding factors (please see the following comments for the discussion section)

Discussion:

Lines 205-207: Authors report as limitations only "Third, other potential confounding variables for symptoms of dry eye and eye fatigue, such as smoking and work stress, are not included in the multivariable logistic regression model due to lack of information". They completely forgot two of the most important factors possibly inducing dry eye symptoms:

1) the first one is especially important as it is an occupational risk typical of the elctonic industry, working with small components: in this sector, very high rates of occupational eye injuries have been reported, including ocular trauma due to the penetration of small components into the eye and due to the irritations caused by the dusts; this issue must be mentioned in the manuscript. Please see and cite e.g. 

- Cai, M.; Zhang, J. Epidemiological Characteristics of Work-Related Ocular Trauma in Southwest Region of China. Int. J. Environ. Res. Public Health 2015, 12, 9864–9875. - Gobba F, Dall'Olio E, Modenese A, De Maria M, Campi L, Cavallini GM. Work-Related Eye Injuries: A Relevant Health Problem. Main Epidemiological Data from a Highly-Industrialized Area of Northern Italy. Int J Environ Res Public Health. 2017 Jun 6;14(6). pii: E604. doi: 10.3390/ijerph14060604. - Yu, T.S.I.; Liu, H.J.; Hui, K. A Case–Control Study of Eye Injuries in the Workplace in Hong Kong. Ophthalmology 2004, 111, 70–74

2) the second important confouder is a factor that, in this case, is not related to occupational exposure, but considering that in Taiwan the average annual UV index is quite high, there is a significant risk for general public of dry eye symptoms due to a very frequent solar UV related disease, that is pterygium, typically causing irritation, dryness and redness of the eyes. This ocular pathology is quite frequent also among young people, in case they spend quite a lot of time outdoor for work or during their leisure time. Prevalence rates up to 20-40% may be observed in this region, and this may explain many of the symptoms: accordingly, solar radiation ocular exposure should be considered in the evaluation of dry eye symptoms. Please see and cite:

Chen, C.L.; Lai, C.H.; Wu, P.L.; Wu, P.C.; Chou, T.H.; Weng, H.H. The epidemiology of patients with pterygium in southern Taiwanese adults: TheChiayi survey. Taiwan J. Ophthalmol. 2013, 3, 58–61.

- Modenese A, Gobba F. Occupational Exposure to Solar Radiation at Different Latitudes and Pterygium: A Systematic Review of the Last 10 Years of Scientific Literature. Int J Environ Res Public Health. 2017 Dec 26;15(1). pii: E37. doi:10.3390/ijerph15010037.

Conclusions

Line 214: Authors can not mention "increased risk", but only positive association

References

The number of references is quite low, and the references are quite old. I suggested some recent references to add, but some more can be included. 

Author Response

Point 1: Abstract session: Please re-write the sentence: "evidence suggests an association between lighting and eye symptoms among cleanroom workers in the electronics industry". "Evidence suggests" means nothing. Furthermore, I am not sure there is so much scientific literature on cleanroom workers, and the association for eye symptoms is not with lighting, but perhaps with insufficient lighting or with specific source of illumination (e.g. white/blu light, etc).

Response 1: Thank you for the comment. Because Su et al. found that flashing light and low illumination of the work environment might be factors of the high prevalence of tear secretion dysfunction and eye symptoms in the electronic enterprise [See reference 4], we have deleted "Evidence suggests" and revised the statement in the Abstract. Please see lines 19-21 of the revised file.

Point 2: Introduction: Line 35-36: please find some more recent data: Authors are presenting data on occupation in the electronic industry related to approximately 10 years ago.

Response 2: Thank you for the comment. The data we described on occupation in the electronics industry is from the International Labour Organization (ILO) survey in 2010 which was released in the 2014 ILO document. Although we searched all important websites and databases again for information related to the global electronics industry, we could not find a more up-to-date global number and could only find data on individual countries. Thus, we cited a recent data on the growth rates of the global electronics industry to help the readers understand the scale of the industry in the Introduction. Please see line 37-38 of the revised file.

Point 3: Introduction: Line 39: I suggest to (prevent) REDUCE dust or lint EXPOSURE.

Response 3: Thank you for the comment. We have revised the statement according to your suggestion in the Introduction. Please see line 42 of the revised file.

Point 4: Introduction: Line 41: High prevalence of dry eye SYMPTOMS.

Response 4: Thank you for the comment. We have revised the statement according to your suggestion in the Introduction. Please see line 44 of the revised file.

Point 5: Materials and Methods: Lines 70-71: some generic indications on the Country where the study was performed are required here, and I'd like to have more occupational details on the type of activity of the electronic enterprise to have a better idea of the work risks(types of materials produced, etc).

Response 5: Thank you for the comment. We have added the information where the study was performed. In addition, the manufactured products and the type of activity were also added in the Materials and Methods. Please see line 79-80 of the revised file.

Point 6: Materials and Methods: "and a final of 91 participants (response rate was 67.4%) remained in the study population": this is a data for the results section.

Response 6: Thank you for the comment. We have moved the sentence from the Materials and Methods section to the Results section. Please see line 122 of the revised file.

Point 7: Results: I have no specific observations on the results of the study, that are almost a description of the eye concerns of the workers with respect to the workplace illumination and occupational activity, but these data are not sufficient to infere any conclusion on possible causal association as there is an inadequate consideration of possible confounding factors (please see the following comments for the discussion section)

Response 7: Thank you for the comment. We completely agree with your comment. Because possible confounding factors were not comprehensively considered in the present study, the results cannot infer any conclusion on possible causal association. Thus, we have revised the statements on data interpretation to avoid any inference on causal association. Please see line 154-155 of the revised file. Furthermore, we have added two important confounding factors-‘occupational eye injuries and solar radiation’ in the limitations of the Discussion section, which were not considered in the present study. Please see line 218-221 of the revised file.

Point 8: Discussion: Lines 205-207: Authors report as limitations only "Third, other potential confounding variables for symptoms of dry eye and eye fatigue, such as smoking and work stress, are not included in the multivariable logistic regression model due to lack of information". They completely forgot two of the most important factors possibly inducing dry eye symptoms:

1) the first one is especially important as it is an occupational risk typical of the elctonic industry, working with small components: in this sector, very high rates of occupational eye injuries have been reported, including ocular trauma due to the penetration of small components into the eye and due to the irritations caused by the dusts; this issue must be mentioned in the manuscript. Please see and cite e.g.

- Cai, M.; Zhang, J. Epidemiological Characteristics of Work-Related Ocular Trauma in Southwest Region of China. Int. J. Environ. Res. Public Health 2015, 12, 9864–9875. - Gobba F, Dall'Olio E, Modenese A, De Maria M, Campi L, Cavallini GM. Work-Related Eye Injuries: A Relevant Health Problem. Main Epidemiological Data from a Highly-Industrialized Area of Northern Italy. Int J Environ Res Public Health. 2017 Jun 6;14(6). pii: E604. doi: 10.3390/ijerph14060604. - Yu, T.S.I.; Liu, H.J.; Hui, K. A Case–Control Study of Eye Injuries in the Workplace in Hong Kong. Ophthalmology 2004, 111, 70–74

2) the second important confouder is a factor that, in this case, is not related to occupational exposure, but considering that in Taiwan the average annual UV index is quite high, there is a significant risk for general public of dry eye symptoms due to a very frequent solar UV related disease, that is pterygium, typically causing irritation, dryness and redness of the eyes. This ocular pathology is quite frequent also among young people, in case they spend quite a lot of time outdoor for work or during their leisure time. Prevalence rates up to 20-40% may be observed in this region, and this may explain many of the symptoms: accordingly, solar radiation ocular exposure should be considered in the evaluation of dry eye symptoms. Please see and cite:

- Chen, C.L.; Lai, C.H.; Wu, P.L.; Wu, P.C.; Chou, T.H.; Weng, H.H. The epidemiology of patients with pterygium in southern Taiwanese adults: TheChiayi survey. Taiwan J. Ophthalmol. 2013, 3, 58–61.

- Modenese A, Gobba F. Occupational Exposure to Solar Radiation at Different Latitudes and Pterygium: A Systematic Review of the Last 10 Years of Scientific Literature. Int J Environ Res Public Health. 2017 Dec 26;15(1). pii: E37. doi:10.3390/ijerph15010037.

Response 8: Thank you for the comment. We agree that occupational eye injuries (e.g. ocular trauma due to small objects or chemical agent) and solar radiation are two important factors possibly inducing dry eye symptoms. We have added these two factors and related statements into our discussion on the limitations in the Discussion. Please see line 218-221 of the revised file.

Point 9: Conclusions: Line 214: Authors can not mention "increased risk", but only positive association.

Response 9: Thank you for the comment. We have changed ‘increased risk’ to ‘positively associated’ in the sentence in the Conclusion. Please see line 230 of the revised file.

Point 10: References: The number of references is quite low, and the references are quite old. I suggested some recent references to add, but some more can be included.

Response 10: Thank you for the comment. We have increased the number of references from 23 to 36 references by adding recent references that are published within 5 years. Please see line 251-326 of the revised file.